# Design and Characterization of a Minimally Invasive Bipolar Electrode for Electroporation

**DOI:** 10.3390/biology9090303

**Published:** 2020-09-21

**Authors:** Giulia Merola, Roberta Fusco, Elio Di Bernardo, Valeria D’Alessio, Francesco Izzo, Vincenza Granata, Deyanira Contartese, Matteo Cadossi, Alberto Audenino, Giacomo Perazzolo Gallo

**Affiliations:** 1Oncology Medical and Research & Development Division, Igea SpA, 41012 Carpi, Italy; giulia.merola13@gmail.com (G.M.); e.dibernardo@igeamedical.com (E.D.B.); v.dalessio@igeamedical.com (V.D.); m.cadossi@igeamedical.com (M.C.); g.perazzolo@igeamedical.com (G.P.G.); 2Hepatobiliary Surgical Oncology Unit, “Istituto Nazionale Tumori IRCCS Fondazione Pascale—IRCCS di Napoli”, 80131 Naples, Italy; f.izzo@istitutotumori.na.it; 3Radiology Unit, “Istituto Nazionale Tumori IRCCS Fondazione Pascale—IRCCS di Napoli”, 80131 Naples, Italy; v.granata@istitutotumori.na.it; 4Complex Structure of Surgical Sciences and Technologies, IRCCS Istituto Ortopedico Rizzoli, Via di Barbiano 1/10, 40136 Bologna, Italy; deyanira.contartese@ior.it; 5Department of Mechanical and Aerospace Engineering, Politecnico di Torino, 10138 Turin, Italy; alberto.audenino@polito.it

**Keywords:** electroporation, bipolar electrode, minimally invasive

## Abstract

**Simple Summary:**

The objective of this study was to test a new bipolar electrode for electroporation consisting of a single minimally invasive electrode. The volume of ablated area is mainly influenced by applied voltage, while the diameter of the electrode had a less impact, making the goal of mini-invasiveness possible. The minimally invasive bipolar electrode is able to treat an electroporated volume of about 10 mm in diameter by using a single-needle minimally invasive electrode.

**Abstract:**

Objective: To test a new bipolar electrode for electroporation consisting of a single minimally invasive needle. Methods: A theoretical study was performed by using Comsol Multiphysics^®^ software. The prototypes of electrode have been tested on potatoes and pigs, adopting an irreversible electroporation protocol. Different applied voltages and different geometries of bipolar electrode prototype have been evaluated. Results: Simulations and pre-clinical tests have shown that the volume of ablated area is mainly influenced by applied voltage, while the diameter of the electrode had a lesser impact, making the goal of minimal-invasiveness possible. The conductive pole’s length determined an increase of electroporated volume, while the insulated pole length inversely affects the electroporated volume size and shape; when the insulated pole length decreases, a more regular shape of the electric field is obtained. Moreover, the geometry of the electrode determined a different shape of the electroporated volume. A parenchymal damage in the liver of pigs due to irreversible electroporation protocol was observed. Conclusion: The minimally invasive bipolar electrode is able to treat an electroporated volume of about 10 mm in diameter by using a single-needle electrode. Moreover, the geometry and the electric characteristics can be selected to produce ellipsoidal ablation volumes.

## 1. Introduction

Electroporation (EP) is a technique developed around 1970 in the biotechnology and medicine fields in order to create nanometer-diameter pores on the cells’ membrane. When cells are exposed to intense pulsed electric fields of at least 400 V/cm as local value [1,2,3], the permeability and conductivity of their membranes increase, allowing local electric field spreading in the cells and their surroundings. The pores on the cells’ membrane permit the entry of substances that otherwise could not enter the cytoplasm [1,2,3,4,5]. As is known, it is possible to use different electroporation parameters (number and voltage amplitude of pulses) in order to obtain a reversible or irreversible effect on the electroporated cells’ membranes.

Irreversible electroporation (IRE) is a non-thermal treatment that uses short electric pulses at high intensity with the aim of determining an electric field above the reversible EP threshold and that applies to the tissue a number of pulses at least 10 times those used for reversible EP [4]. The main effect to tissues due to an IRE treatment is cells necrosis, which can be successfully exploited to treat both cancers [6,7,8] and non-cancerous pathologies [9,10,11].

In the reversible electroporation, pulses’ amplitude and duration are such as to allow pore resealing, preserving the cells’ viability after the pulse delivery. Reversible electroporation is largely used in medicine, where EP can be combined with an anticancer agent (electrochemotherapy, or ECT) or can facilitate the transferring of a DNA plasmid into cells (electro-gene-transfer, or EGT) [12,13,14,15,16,17,18].

ECT is used for the treatment of cutaneous and subcutaneous tumors of different histotypes, such as skin and non-skin cancers, as well as metastases and recurrences. More recently, ECT has begun to replace thermal ablation techniques in the treatment of deep tumors [19,20,21,22,23,24,25,26,27,28,29,30,31,32]. New deployable and expandable electrode prototypes have been designed and tested for laparoscopic applications on the liver and for endoscopic trans-oral and trans-anal procedures. Electroporation treatment using new deployable expandable electrode prototypes is safe and feasible, allowing a gradual increase in the ablated area by segmenting the electroporation in several steps [33].

The aim of the study was to design and characterize a minimally invasive bipolar electrode for electroporation. The new device is different from the already marketed electrodes for its geometry, design and utilization. It consists of a single bipolar electrode, whereby the anode and cathode components are contained on the same needle. The new bipolar electrode has been explored as an approach to reduce the number of needles required to be inserted, which may substantially simplify the electrode placement procedure, minimize invasiveness and save time. In fact, the already marketed electrodes involve the use of two different needles, as opposite poles, and require the correct needle placement to be guaranteed.

The device was realized to improve and facilitate the application of electroporation and is aimed to be connected to the electroporator device which controls the generation and delivery of electrical impulses and evaluates the procedure outcome. The intended purpose of the new electrode is to reach a trade-off among the treatment invasiveness, linked to the probe diameter, and the electroporated area. Therefore, we tried to reduce as much as possible the diameter of the device while still ensuring an electroporated area of at least 10 mm of diameter.

Based on this purpose, different geometric aspects, such as the diameter of the electrode, the length of the conductive and insulated poles and the applied voltage, were analyzed. Furthermore, the ablation area obtained with the bipolar electrode prototype was evaluated in pre-clinical tests.

The work consists of three main phases: 1: Description of the electrode design, 2: description of the theoretical study carried out with the Comsol Multiphysics^®^ software [34] and 3: analysis of both vegetables and pig pre-clinical experimental tests.

## 2. Results

### 2.1. Theoretical Study Results

#### 2.1.1. Symmetric Geometry

The first test (Table 1) analyzed how the change in the electrode diameter (1.40–1.50–1.60–1.80–2.00 mm) influences the electric field distribution, while the lengths of poles and applied voltage were not changed. The chosen electrode diameter is within range of already commercialized needles (VGD needles series of Igea SpA, Carpi-Modena, Italy) that vary among 0.8–1.8 mm.

Small variations of the electric field around the electrode have been observed (Figure 1a,b and Figure 2a). A minimal increase of the electric field was measured by increasing the electrode diameter. The increase of the diameter from 1.40 to 1.50 mm did not determine an increase of the electroporated volume around the conductive poles P1 and P2. On the contrary, with a diameter of 1.60 mm, the electroporated volume increased by 5%, and with diameters of 1.80 and 2.00 mm, the electroporated volume increased by 10%. Around the insulating pole S, there was not an increase in the electroporated volume for the diameters of 1.40, 1.50 and 1.60 mm, while for diameters of 1.80 and 2.00 mm, the volume increased by 11%. For this reason, the smallest diameter implemented (1.40 mm) has been fixed for designing the electrode prototype, satisfying the minimum invasive requirements.

The second test session (Table 1) analyzed how a change in the applied voltage (500–1000–1500 V) influences the electric field distribution, while the lengths of poles and the diameter were kept unchanged. As the voltage increased from 500 to 1000 V, the electroporated volume also increased in diameter (Figure 1c,d and Figure 2b): along the P1 and P2 poles, the electroporated volume diameter increased from 8 to 10 mm (25%), and along the insulated spacer S, the electroporated volume diameter increased from 7 to 9 mm (29%). Applying a voltage of 1500 V, the diameter of the electroporated volume increased by 75% around the P1 and P2 conductive poles, while the increase was 86% around the insulating pole S.

The third test session (Table 1) analyzed the influence of the lengths of the insulated pole (3.00–5.00–7.00 mm) on the electric field distribution, while the applied voltage, the electrode diameter and the conductive poles length were unchanged. The simulation showed that an increase in the insulated pole (S) length causes a decrease in the electric field distribution around the pole itself (Figure 1e,f and Figure 2c): varying the length of S from 3 to 5 mm, a decrease of 11% was observed (from 9 to 8 mm), and when the length of S was equal to 7 mm, the reduction was 22% (7 mm). Therefore, the minimum possible value of insulated pole length (3.00 mm) was fixed.

The last test session (Table 1) allowed the length of the conductive poles (P1 and P2) to be analyzed (3.00–5.00–10.00–15.00 mm). Increasing the length has determined an increase of electroporated volume (Figure 1g,h and Figure 2d). As the length of the conductive poles increased from 3 to 5 mm, an increase of 11% in the diameter of the electroporated volume was observed. The electric field due to the electrode with a 10 mm pole length had a diameter of 12 mm (increase of 33%), while an overall field distribution diameter of up to 14 mm (increase of 56%) was achieved with a 15 mm pole length. A less pronounced increase in the field distribution was also observed around the insulation pole (increase of 22% with a 10 mm conductive pole length, increase of 44% with a 15 mm conductive pole length).

#### 2.1.2. Asymmetric Geometry

In the second part of the theoretical study, the same variables used in the analysis on electrodes with symmetric geometries were analyzed (Figure 3 and Figure 4).

The test session 5 (Table 1) showed that the increase of the applied voltage determines an irregular increase in the electroporated volume, or rather, the increase due to the short pole is greater than the increase due to the long one (Figure 3a,b and Figure 4a). As the voltage increased from 500 to 800 V, the electroporated volume around the P1 conductive pole increased from 12 to 14 mm (17% increase), while around the P2 conductive pole it increased from 8 to 10 mm (25%). Around the insulating pole S, the electroporated volume diameter increased from 10 to 12 mm (20%).

The test session 6 (Table 1) analyzed both the change in diameter and voltage. While changing the voltage from 500 to 800 V influenced the electric field distribution, no change was obtained due to different poles diameters (Figure 3c,d and Figure 4b). The change in voltage showed an increase in the electric field diameter around the P1 conductive pole from 12 to 14 mm (17%) and from 10 to 12 mm (20%) around the insulating pole S; the maximum increase, however, occurred around the P2 conductive pole (25%).

The test session 7 (Table 1) showed that an increase in the insulated pole length causes a decrease in the electric field distributed around the pole itself (decreased by 50%). The change in length, instead, does not alter the field distribution around conductive poles (Figure 3e,f and Figure 4c).

The test session 8 (Table 1) showed that the electric field observed around the short conductive pole is larger than the electric field observed around the long one (Figure 3g,h and Figure 4d). As can be seen from the data, the greater the difference between the poles’ lengths, the greater the irregularity of the electrical field distribution. The largest field diameter was obtained for a pole length of 5 mm, regardless of the pole (P1 or P2): the field had an extension four times greater than that of the field due to the 20 mm long pole.

### 2.2. Vegetable Model Results

During the pre-clinical study on a vegetable model, 14 tests have been conducted on potatoes (Table 2).

The black area, visible in the Figure 5, represents the electroporated volume observed 24 h after the treatment and due to the irreversible electroporation. The delay of 24 h was chosen to better visualize the ablated area, not visible immediately after the electroporation ended.

By fixing the poles’ length (P1, S and P2) and changing the applied voltage, the greater electroporated volume occurred at the higher voltage and around the pole with the shorter length (Tests 13 and 14, Table 2). As voltage decreased, the diameter of the electroporated volume around P1 and P2 conductive poles decreased from 18 to 12 mm (decrease of 31%) and from 13 to 10 mm (decrease of 18%), respectively. Around the insulating pole S, there was a decrease from 19 to 12 mm (35%).

By fixing the insulated pole length and the applied voltage, a larger diameter of the electric field was obtained in the case of the electrode with a longer total length (Tests 6 and 10, Table 2). The results showed that the electroporated volume diameter around the conductive poles P1 and P2 decreased, in both cases, by 32%, while around the insulating pole S, there was a decrease of 47%.

By fixing the total length (P1 + S + P2) and the applied voltage, the prototype with symmetric geometry determined a more homogeneous and regular electric field distribution than the electrode with asymmetrical geometry (Tests 1 and 8, Table 2).

The last aspect analyzed was the influence of the spacer length. Tests showed that an increase of the electroporated volume around the spacer is observable when the pole length itself decreased (Tests 4 and 14, Table 2). The insulating pole S decreasing, from 5 to 3 mm, determined an increase in the diameter of the electroporated volume around S from 12 to 15 mm (19%).

### 2.3. Animal Model Results

The histological results on the liver specimens highlighted the difference between the lighter and the darker area on the electroporated tissue and therefore, the living cells from the dead ones. The analysis of the ablated areas (dark areas of each specimen) showed that an electroporated area with a diameter of about 10 mm can be obtained (Table 3, Figure 6). In fact, bipolar electrode prototype, tested in open surgery, determined a parenchymal damage area, due to irreversible electroporation, with a maximum diameter higher than 9 mm in all experiments. The largest damage area was obtained applying ten 8-pulse trains at 800 V and using a bipolar electrode characterized by a distal pole of 5.00 mm, an insulated spacer of 3.00 mm and a proximal pole of 10.00 mm (Test 3, Table 3). Caspase staining showed that positive area was higher than the ablated one (data not shown).

## 3. Discussions and Conclusions

IRE and ECT procedures are influenced by the physical and electric conditions of the treatment, including electrode geometry, applied electric parameters and local tissue properties [35,36].

Thus far, initial optimization studies and the current standard treatment are predicated on precise placement of multiple needles in an array, with each respective electrode pair spaced approximately 1–3 cm apart. This potentially is a time-consuming approach; additionally, the current practice of accurately targeting and precisely aligning an array of multiple electrodes can prove to be a cumbersome task in clinical practice. Needles design, such as a single-insertion bipolar electrode, whereby the anode and cathode components are contained on a single needle, have been explored as an approach to reduce the number of electrode insertions required [35,36,37,38,39], which may substantially simplify the electrode placement procedure, saving time and increasing accuracy.

Wandel et al. [35] investigated a bipolar electrode for IRE treatments with the goals of characterizing, optimizing and evaluating the physical and electric conditions’ effects on the size of treatment zones and determining reproducibility of the results. This study confirms the potential utility of using a paradigm of IRE treatment based on a single-insertion applicator bipolar electrode configuration. Their results show that application of energy via bipolar IRE electrodes corresponds to previous observations noted for monopolar systems—that increases in voltage and number of cycles increase ablation size. Bipolar IRE ablation zones can be increased with repetitive high voltage and greater pulse widths. In their study, they did not analyze the ablation area in function of asymmetrical configuration as in this study. Asymmetric geometry has been investigated in order to evaluate the capability of shaping the electroporated area. Thanks to the asymmetric geometry, in fact, it is possible to focus the electric field distribution around the shortest pole more than around the longest one. In this way, we obtain a “drop”-shaped electroporated volume, which can be used to treat specific cancer lesions (such as the vertebral metastasis), preserving tissue not intended to be treated.

Bipolar electrodes were also realized for radiofrequency ablation (RFA) via the endoscopic approach to treat inoperable malignant tumor in biliary structures, avoiding stents occlusion, ablating ingrowth of blocked metal stents, prolonging stents patency and ablating residual adenomatous tissue after endoscopic ampullectomy [40,41,42,43].

Microwave ablation (MWA) has emerged as a minimally invasive therapeutic modality and is used for treating unresectable tumors and in other applications. Components of image-guided MW ablation systems include: high-power MW sources, ablation applicators that deliver power from the generator to the target tissue, cooling systems, energy-delivery control algorithms and imaging guidance systems tailored to specific clinical indications. The applicator incorporates a MW antenna that radiates MW power into the surrounding tissue. A variety of single antenna designs have been developed for MW ablation with the objective of efficiently transferring MW power to tissue, with a radiation pattern well matched to the size and shape of the targeted tissue [44,45,46,47].

Instead, no bipolar electrode was available for ECT treatments; therefore, the aim of our study was to design and characterize a minimally invasive bipolar electrode for reversible electroporation. The main goal of our study was to reach a trade-off among the treatment invasiveness, linked to the probe diameter, and the electroporated area. Unlike the conventional microwave (MWA) single probes, which can reach ablated area greater than 10 mm despite the invasiveness (diameter are much higher than 1.4 mm [44,45,46,47]), we tried to reduce as much as possible the diameter of the device while still ensuring an electroporated area of at least 10 mm of dimeter. In fact, as reported by Fallahi et al. [45], although printed circuit board technologies such as microstrips provide the freedom to design antennas with diverse ablation patterns, achieving a practical device with an overall diameter less than 3 mm is not feasible at common MW ablation frequencies of 915 MHz and 2.45 GHz due to the large wavelength. In Colebeck et al. [47], a slot antenna was printed on the back side of a microstrip line with ultra-wideband characteristic. The device is capable of creating ablation zones at 915 MHz, 2.45 GHz and 5.8 GHz. However, the overall width of the device is 5.5 mm, making it unsuitable for most clinical MW ablation applications.

Our simulation findings demonstrate that the applied voltage is the main variable to influence the electroporated volume. The maximum voltage, chosen in the experiments, of 1500 V, is linked to the electric field (voltage on distance between two poles) and it is sufficient to reach the irreversible electroporation threshold. Moreover, a higher voltage value should be avoided to prevent electric hazards (sparks, high current, etc.).

Furthermore, changing both the conductive and insulated poles’ lengths affected the distribution of the electric field. The conductive pole’s length determined an increase of electroporated volume, while the insulated pole’s length inversely affects the electroporated volume size and shape; when the insulated pole length decreases, a more regular shape of the electric field is obtained. The electroporated volume has a more irregular shape (“drop” shape) compared to the one due to the electrode with symmetric geometry. The tests performed on vegetables are in accordance with the simulations, suggesting that the voltage and the total length of the electrode are the variables that mostly influence the electroporated volume. Tests performed on a pig pre-clinical model with bipolar electrode prototypes showed that a parenchymal damage could be obtained, due to irreversible electroporation protocol, in an area with a maximum diameter higher than 9 mm in all experiments.

The largest damage area was obtained with 80 pulses at 800 V of applied voltage and a bipolar electrode characterized by a distal pole of 5.00 mm, an insulated spacer of 3.00 mm and a proximal pole of 10.00 mm.

In conclusion, using a bipolar electrode, it is possible to simultaneously reduce the invasiveness of the treatment and guarantee an electroporated volume of about 10 mm in diameter. Its less invasiveness makes the bipolar electrode particularly suitable in electroporation-based treatments of deep-seated tumors of limited size. The size of electroporation area of about 10 mm can be considered a limitation of the device; however, multiple insertion and treatments could be performed in case of lager lesions.

Moreover, the bipolar electrode can allow to treat metastases localized to the vertebrae with a minimally invasive percutaneous approach through the vertebral peduncle. A fast treatment compared to thermo- and cryo-ablation which also leaves the regenerative capacity of the bone tissue intact.

## 4. Materials and Methods

### 4.1. Electrode Configuration

To develop a minimally invasive electrode, we realized an alternating concentric tube of insulated and conductive material, with both positive and negative poles on the same needle.

Austenitic stainless-steel number 304 (according to the American Iron and Steel Institute (AISI) system) was selected for the electrode conductive poles because this material has good mechanical properties (tensile strength, yield strength, high corrosion resistance) compared to ferritic and martensitic steels, and it also guarantees longevity of the material, chemical stability and corrosion resistance.

For the insulating pole, the choice of materials fell between polyimide and polyethylene terephthalate, being among the most used polymers in the biomedical field. Both are thermoplastic polymers. Polyimide (PI) was chosen for its coefficient of friction, good vibration-damping properties, abrasion resistance and high dielectric strength.

As shown in Figure 7, an inner stainless-steel needle with Trocar tip was used to make the proximal conductive pole (P2). A hollow concentric PI tube is inserted above the inner needle to create the insulated spacer (S), then a concentric outer stainless-steel tube is placed to make the distal conductive pole (P1); finally, a concentric PI tube makes the outer insulated sheath.

A more simplified electrode geometry has been used for Comsol Multiphysics^®^ simulations (Figure 8), consisting of two conductive poles and an insulated spacer.

### 4.2. Comsol Multiphysics^®^

Comsol Multiphysics^®^ modelling software was used to identify the best solutions for the geometric and electrical characterization of the device [34] by using the Finite Element Method (FEM) algorithm. The software allows to reproduce the electrode geometry and also analyze the component materials and their effect on the electric field distribution. Stainless-steel AISI 4340 was chosen for the electrode conductive poles since its properties are similar to AISI 304 and it is available in the Comsol Multiphysics^®^ database (Table 4). Polyimide properties are shown in Table 5. During the simulation, animal liver properties were considered (Table 6). The simulated tissue/organ were represented as a cube in which to immerse the device. We have set the voltage applied as static and the mesh as unstructured with tetrahedral elements. The computation elements used to recreate the model are full cylinders for two conductive poles and the insulating pole.

### 4.3. Theoretical Study

The analysis was carried out in three steps involving the study of different geometric and electrical parameters. The following parameters were analyzed: electrode diameter, applied voltage through the conductive poles, insulated spacer length (S) and conductive poles’ (P1 and P2) length. These geometric parameters were analyzed throughout several combinations (see Table 1).

Tests were divided in two main sections with the aim of investigating the effects of the electrode geometry, differentiating between symmetric and asymmetric geometry. In the first case, the conductive poles have the same length (test session 1–4 of Table 1); in the other case, the conductive poles have a different length (test session 5–8 of Table 1).

In the theoretical study, the electric field around the electrode, due to the applied voltage, was visible on the “yz” plane. The tests were conducted with a threshold of minimum 400 V/cm. The electric field studied would reach values between 350 × 10^2^ V/cm and 400 × 10^2^ V/cm.

### 4.4. Prototype Design

Electrode prototypes with an overall diameter of 1.40 mm were realized. To avoid the creation of positive steps, a concentric structure of insulated and conductive materials characterized by minimum wall thicknesses was developed and an adequate mechanical machining was designed. The stainless-steel mandrel has an outer diameter of about 0.80 mm, except for the section intended to be the proximal conductive pole (P2), whose outer diameter is of 1.40 mm. The Trocar tip was made in the end part of the mandrel and its shape helps the insertion and drilling of the target organ. Then, the first insulated sheath, the stainless-steel tube making the distal conductive pole (P1), and the second insulated sheath are placed (in a concentric way) on the mandrel section where the diameter is 0.80 mm. As already highlighted in the study of the material, the thickness of the insulated sheath has to guarantee the electrical insulation for all the allowed applied voltages (with a maximum functional voltage of 3000 V). For the realization of these prototypes, the material discussed in Section 2.1 was chosen, as it satisfies the dielectric requirements. The geometric parameters of the device are shown in Table 7.

The prototypes made in Igea SpA were used for pre-clinical studies on both vegetable (potatoes) and animal (pigs) models.

### 4.5. Vegetable Model

Potatoes of the “Agata CAT.I.CAL.40” variety produced by Covone Domenico S.r.l. (Aquila, Abbruzzo, Italy) are used in this study. The pre-clinical study on a vegetable model has been performed on potatoes, thanks to their produced electric field which is very similar to that observed in both animal and humans organs/tissues [5,35]. In each test, the electrode was inserted into the potato for all its length, making sure that the conductive part was at least 10 mm deep inside. The electrode was then connected to a Cliniporator Vitae^TM^ device, manufactured by Igea SpA (Carpi, Italy). According to the irreversible electroporation protocol, each test was characterized by the following parameters: twelve 8-pulse trains, 100 μs pulse length, 1 Hz pulse frequency and applied voltage between 500 and 1500 V. In order to visualize the electroporated area, a section cutting took place 24 h after each test.

### 4.6. Animal Model

All applicable international, national and/or institutional guidelines for the care and use of animals were followed. The in vivo experiments have been conducted respecting bio-ethic principles and current Italian (art. 31 D.L. 26/2014) and European regulations. The use of the animal model was authorized by the Ministry of Health (n. 15/2019-PR) within the Research Project “Expandable electrode for electroporation of tissues”. This article does not contain any studies with human participants performed by any of the authors. In order to evaluate the bipolar electrode capability to treat the desired volume, two experimental sessions were performed on two anesthetized pigs, using the parameters shown in Table 3. The in vivo experiments were conducted respecting bio-ethic principles and current Italian (art. 31 D.L. 26/2014) and European regulations. The use of the animal model was authorized by the Ministry of Health (n. 15/2019-PR) within the research project named “Expandable electrode for electroporation of tissues”.

The animals (Females Sus Scrofa Large White, 60 kg of weight) were anesthetized according to the following anesthetic/analgesic protocol: zoletil anesthesia 50/50 0.5 mL/kg intramuscular (i.m.) + propofol 6 mg/kg intravenous (i.v.) + ketamine 10 mg/kg i.v. + sevorane 2% by inhalation, and butorphanol analgesia 0.1–0.4 mg/kg i.m. or i.v. At the end of each experiment, the animals were sacrificed and liver specimens were removed for vital staining (Tetrazolium) and/or histology and immunohistochemistry evaluation. The electrode was connected to a Cliniporator Vitae^TM^ device and used in open surgery.

In the tests 1–4 (Table 3), the impact on the electric field distribution due to the applied voltage and the total electrode length—in particular the proximal pole (P2)—was analyzed, using the following protocol: ten 8-pulse trains, applied voltage between 800 and 1200 V and pulse duration of 100 μs. The electrode used had a proximal pole (P2) length between 20.00 (electrode 1) and 10.00 mm (electrode 2). The different protocols were chosen based on simulations and potato experiments.

### 4.7. Histological Analysis

Tissues were macroscopically and histologically examined. The specimens were obtained sectioning the liver along the orthogonal plane. Each sample contains both the region close to the electrode and the surrounding area, and it also includes an adjacent rim of non-ablated hepatic parenchyma. The tissues were processed and embedded in paraffin and the sections obtained were stained with Hematoxylin/Eosin (Sigma Aldrich, Milan, Italy) for subsequent histological evaluations. A mean number of 6 slides per specimen was analyzed (range from 2 to 12 slides). The presence of necrosis (type and extension), inflammatory infiltrates, changes in blood vessels and biliary ducts in both the treated and non-treated areas and eventual residual vital tissue (amount and location) were analyzed. A light microscope (Olympus BX51) was used to observe the slides and the Aperio digital scanner (AperioScanscope CS System, Aperio Technologies, Vista, CA, USA) was used to capture representative images from each slide.

### 4.8. Immunohistochemical Analysis

Additional slides for each specimen were immuno-stained with the caspase-3 (CASP3) method, according to standard protocol [48]. Negative controls, obtained omitting the primary antibody, were included to check proper specificity and performance of the applied reagents.

## Figures and Tables

**Figure 1 biology-09-00303-f001:**
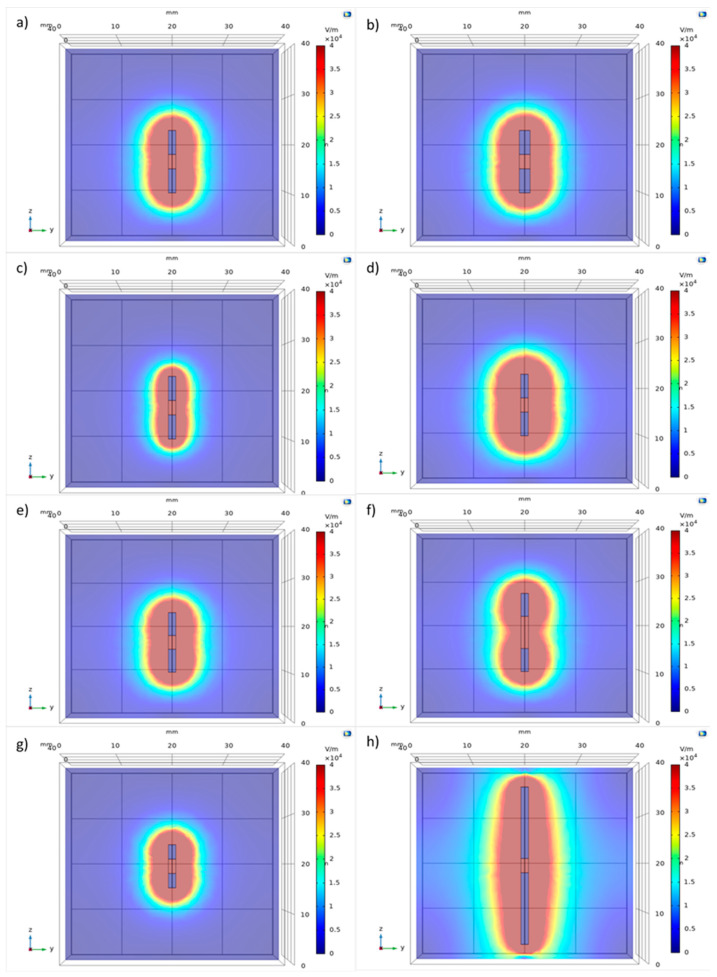
The electric field distribution simulated with Comsol Multiphysics^®^. Field distribution due to an electrode diameter of 1.40 and of 2.0 mm is respectively shown in (**a**,**b**). Field distribution due to an applied voltage of 500 and of 1500 V is respectively shown in (**c**,**d**). Field distribution due to an insulated pole of 3.00 and of 7.00 mm is respectively shown in (**e**,**f**). Field distribution due to a length of both the conductive poles of 3.00 and of 15.00 mm is respectively shown in (**g**,**h**). The color bar represents the intensity of the electric field distribution: in red is the highest value of electric field of the electroporated zone that decreases when moving away from the poles of the electrode.

**Figure 2 biology-09-00303-f002:**
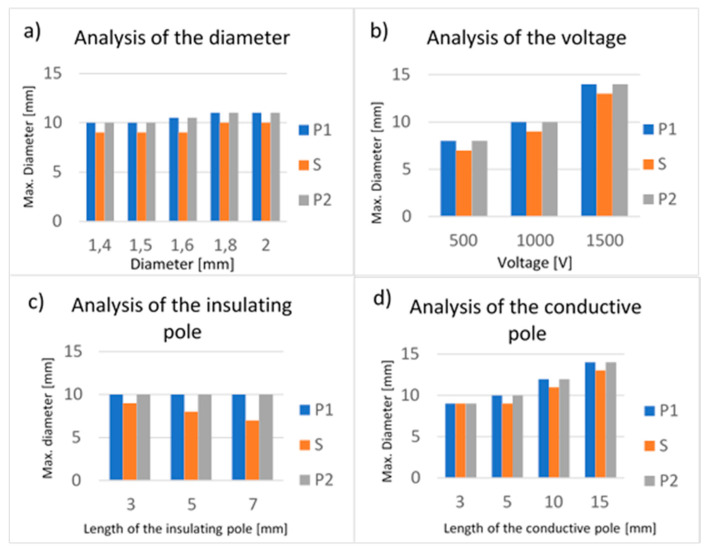
Histograms of tests on electrodes with symmetric geometry (Test 1–4) of Figure 1. In (**a**–**d**) are reported the values of the maximum diameters measured at the distal pole (P1), insulated spacer (S) and proximal pole (P2).

**Figure 3 biology-09-00303-f003:**
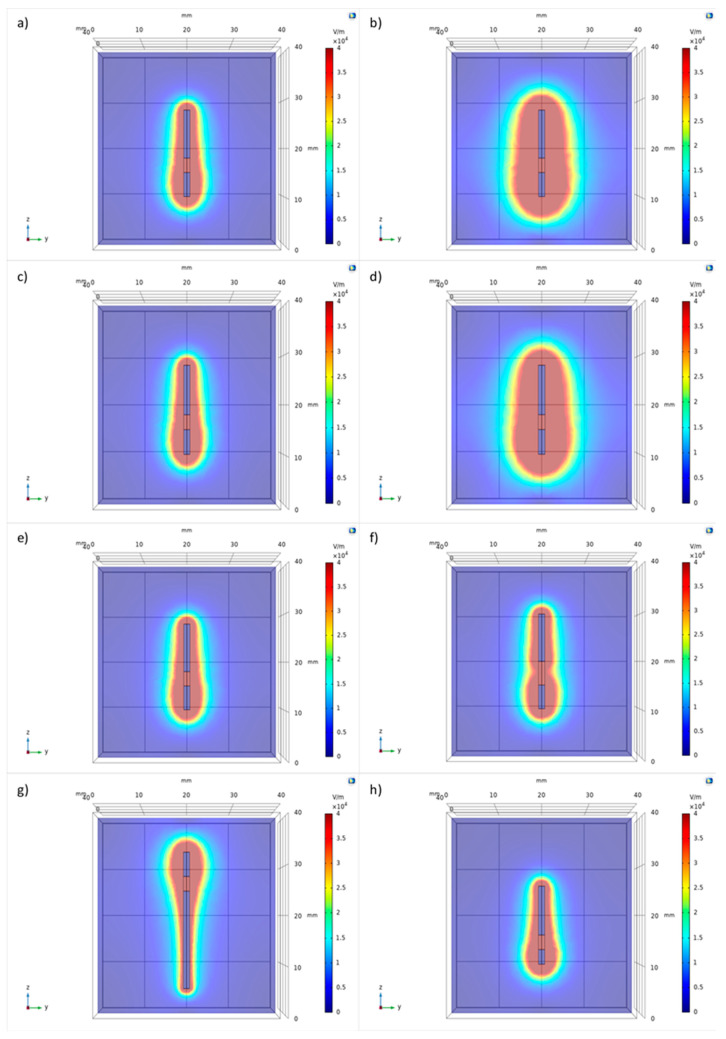
Comsol Multiphysics^®^ simulation: The electric field distribution considering a voltage of 500 V in (**a**) and of 1500 V in (**b**). The electric field distribution due to an electrode’s diameter of 1.40 mm and a voltage of 500 V in (**c**) and a diameter of 1.50 mm and a voltage of 1500 V in (**d**). Insulated pole of 3.0 mm in (**e**) and of 5.0 mm in (**f**). P1 = 20.00 mm and P2 = 5.00 mm in (**g**), P1 = 3.00 mm and P2 = 10.00 mm in (**h**). The color bar represents the intensity of the electric field distribution: in red is the highest value of electric field of the electroporated zone that decreases when moving away from the poles of the electrode.

**Figure 4 biology-09-00303-f004:**
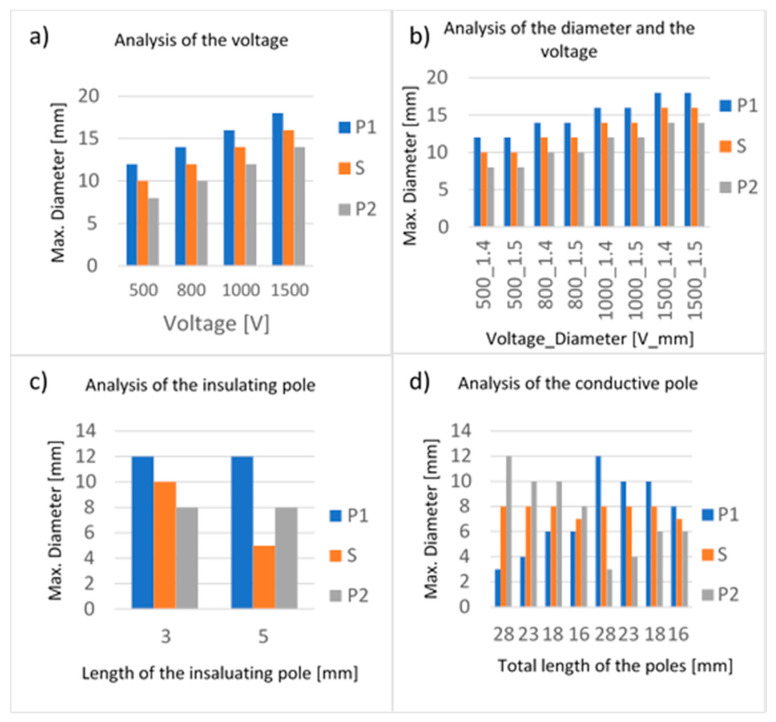
Histograms for tests with electrodes with asymmetric geometry (Test 5–8) of Figure 1. In (**a**–**d**) are reported the values of the maximum diameters measured at the distal pole (P1), spacer (S) and proximal pole (P2).

**Figure 5 biology-09-00303-f005:**
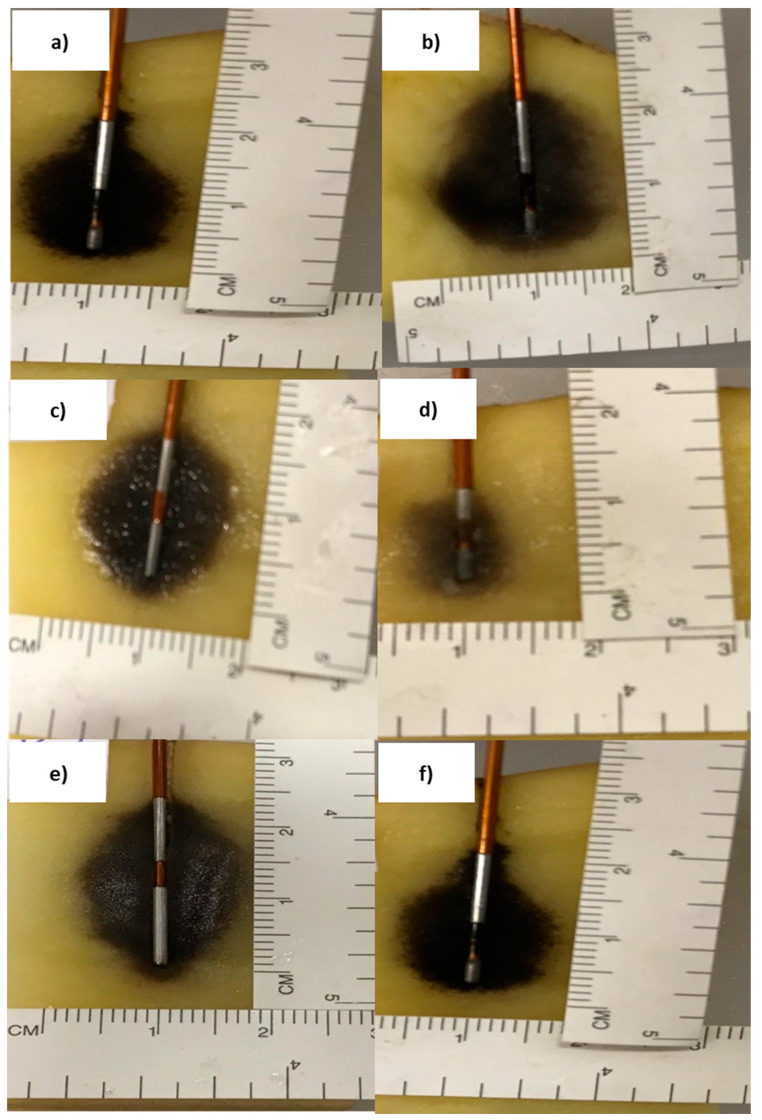
Examples of tests conducted on potatoes of Table 2. The black zone identifies the electroporated area 24 h after the treatment, while the surrounding area is vegetable material that is not electroporated. (**a**) Test 13 (asymmetric geometry) = P1: 3.00 mm, S: 5.00 mm, P2: 10.00 mm, Voltage: 800 V. (**b**) Test 14 (asymmetric geometry) = P1: 3.00 mm, S: 5.00 mm, P2: 10.00 mm, Voltage: 500 V. (**c**) Test 6 (symmetric geometry) = P1: 5.00 mm, S: 3.00 mm, P2: 5.00 mm, Voltage: 500 V. (**d**) Test 10 (symmetric geometry) = P1: 3.00 mm, S: 3.00 mm, P2: 3.00 mm, Voltage: 500 V. (**e**) Test 1 (symmetric geometry) = P1: 10.00 mm, S: 3.00 mm, P2: 10.00 mm, Voltage: 500 V. (**f**) Test 14 (asymmetric geometry) = P1: 3.00 mm, S: 5.00 mm, P2: 10.00 mm, Voltage: 500 V.

**Figure 6 biology-09-00303-f006:**
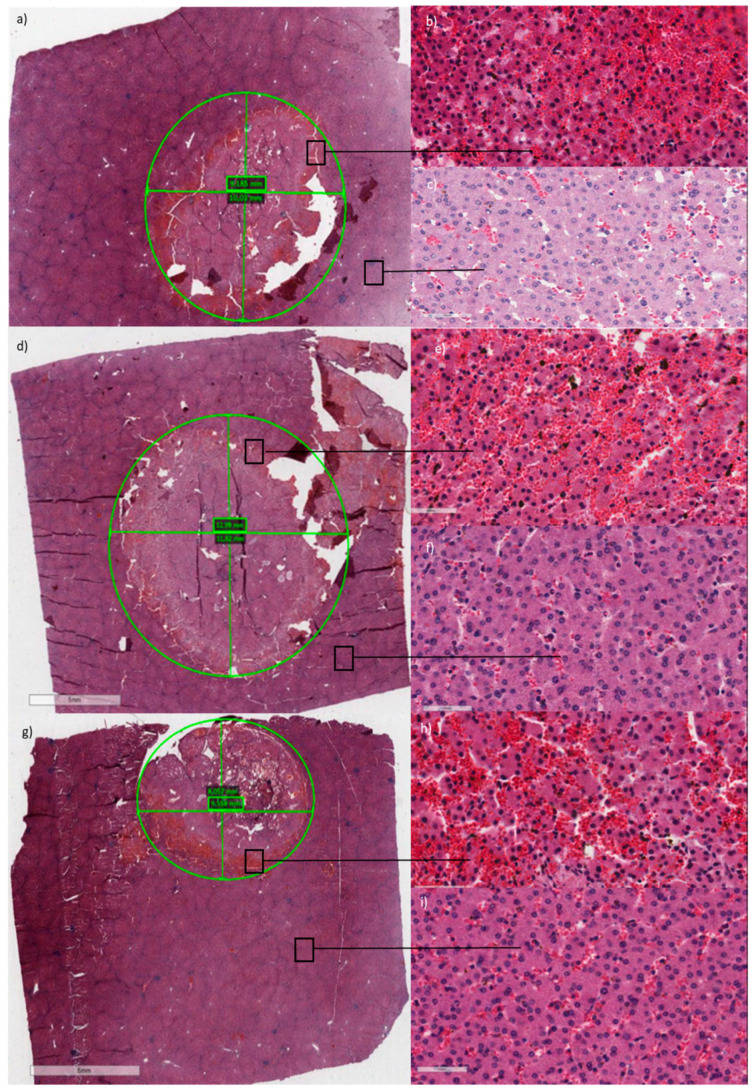
Macroscopic images and histological analysis (hematoxylin and eosin staining) of porcine liver specimens. (**a**) Details of the damaged parenchyma using the setting of Test 1, Table 3, in (**b**) zoom of altered parenchyma, and in (**c**) zoom of normal parenchyma. (**d**) Details of the damaged parenchyma using the setting of Test 3, Table 3, in (**e**) zoom of altered parenchyma, and in (**f**) zoom of normal parenchyma. (**g**) Details of the damaged parenchyma using the setting of Test 4, Table 3, in (**h**) zoom of altered parenchyma, and in (**i**) zoom of normal parenchyma. Red structures in (**b**,**e**,**h**) are capillaries.

**Figure 7 biology-09-00303-f007:**
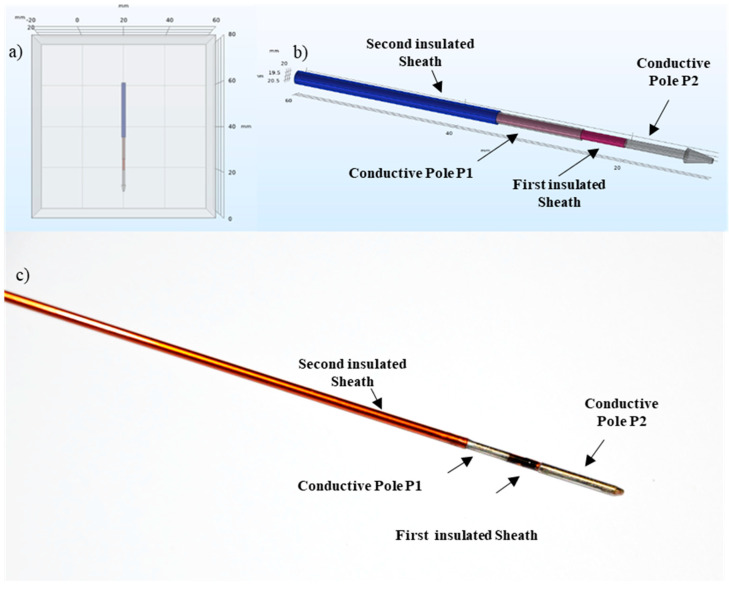
Bipolar coaxial electrode. (**a**) Positioning of the device inside the simulated organ (cube). (**b**) Electrode geometry implemented in Comsol Multiphysics^®^: two conductive poles, a tip and two insulated sheaths. (**c**) An example of bipolar electrode prototype.

**Figure 8 biology-09-00303-f008:**
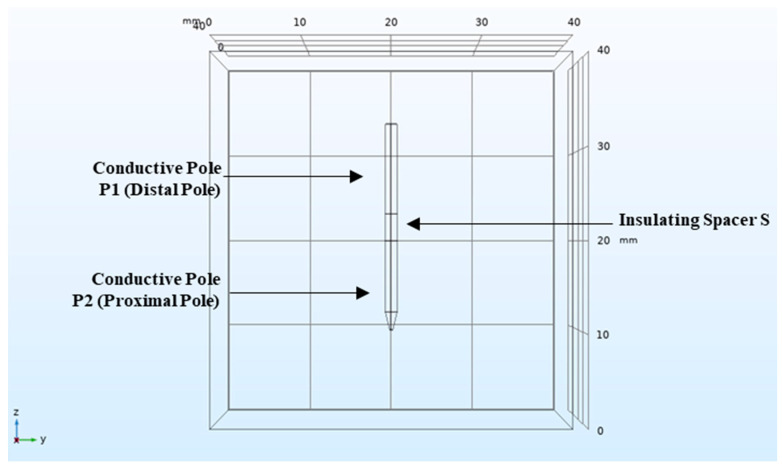
Simplified electrode geometry implemented in Comsol Multiphysics^®^ and used for theoretical study. The geometry is composed of two conductive poles (austenitic stainless-steel) and one insulated spacer (polyimide). The electrode is inserted in a cube representing the organ of interest.

**Table 1 biology-09-00303-t001:** Tests carried out with Comsol Multiphysics^®^ software for the analysis of the electric field distribution. In bold are the values changed in each test. Symmetric geometry is analyzed in tests 1–4, and asymmetric geometry is analyzed in tests 5–8. Values of maximum diameters obtained after electroporation are showed in the last three columns.

Test Session	Changed Variable	Conductive Pole P1 (mm)	Insulated Pole S (mm)	Conductive Pole P2 (mm)	Voltage (V)	Diameter D (mm)	Diameter of Electroporated Volume along P1 (mm)	Diameter of Electroporated Volume along S (mm)	Diameter of Electroporated Volume along P2 (mm)
**1**	Diameter	5.00	3.00	5.00	1000	1.40	10.00	9.00	10.00
		5.00	3.00	5.00	1000	1.50	10.00	9.00	10.00
		5.00	3.00	5.00	1000	1.60	10.50	9.00	10.50
		5.00	3.00	5.00	1000	1.80	11.00	10.00	11.00
		5.00	3.00	5.00	1000	2.00	11.00	10.00	11.00
**2**	Voltage	5.00	3.00	5.00	500	1.40	8.00	7.00	8.00
		5.00	3.00	5.00	1000	1.40	10.00	9.00	10.00
		5.00	3.00	5.00	1500	1.40	14.00	13.00	14.00
**3**	Spacer	5.00	3.00	5.00	1000	1.40	10.00	9.00	10.00
		5.00	5.00	5.00	1000	1.40	10.00	8.00	10.00
		5.00	7.00	5.00	1000	1.40	10.00	7.00	10.00
**4**	Conductive poles	3.00	3.00	3.00	1000	1.40	9.00	9.00	9.00
		5.00	3.00	5.00	1000	1.40	10.00	9.00	10.00
		10.00	3.00	10.00	1000	1.40	12.00	11.00	12.00
		15.00	3.00	15.00	1000	1.40	14.00	13.00	14.00
**5**	Voltage	5.00	3.00	10.00	500	1.40	12.00	10.00	8.00
		5.00	3.00	10.00	800	1.40	14.00	12.00	10.00
		5.00	3.00	10.00	1000	1.40	16.00	14.00	12.00
		5.00	3.00	10.00	1500	1.40	18.00	16.00	14.00
**6**	Voltage and Diameter	5.00	3.00	10.00	500	1.40	12.00	10.00	8.00
		5.00	3.00	10.00	500	1.50	12.00	10.00	8.00
		5.00	3.00	10.00	800	1.40	14.00	12.00	10.00
		5.00	3.00	10.00	800	1.50	14.00	12.00	10.00
		5.00	3.00	10.00	1000	1.40	16.00	14.00	12.00
		5.00	3.00	10.00	1000	1.50	16.00	14.00	12.00
		5.00	3.00	10.00	1500	1.40	18.00	16.00	14.00
		5.00	3.00	10.00	1500	1.50	18.00	16.00	14.00
**7**	Spacer	5.00	3.00	10.00	500	1.40	12.00	10.00	8.00
		5.00	5.00	10.00	500	1.40	12.00	5.00	8.00
**8**	Conductive Poles	20.00	3.00	5.00	500	1.40	3.00	8.00	12.00
		15.00	3.00	5.00	500	1.40	4.00	8.00	10.00
		10.00	3.00	5.00	500	1.40	6.00	8.00	10.00
		10.00	3.00	3.00	500	1.40	6.00	7.00	8.00
		5.00	3.00	20.00	500	1.40	12.00	8.00	3.00
		5.00	3.00	15.00	500	1.40	10.00	8.00	4.00
		5.00	3.00	10.00	500	1.40	10.00	8.00	6.00
		3.00	3.00	10.00	500	1.40	8.00	7.00	6.00

**Table 2 biology-09-00303-t002:** Tests performed during the pre-clinical study on a vegetable model. Total length: P1 + S + P2. The maximum diameters of the electroporated volume around the poles, observed 24 h after the treatment, are also reported.

	P1 (mm)	S (mm)	P2 (mm)	Total Length (mm)	Voltage (V)	Diameter of Electroporated Volume along P1 (mm)	Diameter of Electroporated Volume along S (mm)	Diameter of Electroporated Volume along P2 (mm)
**1**	10.00	3.00	10.00	23.00	500	12.30	14.30	12.10
**2**	10.00	5.00	10.00	25.00	800	15.20	18.20	10.00
**3**	10.00	5.00	10.00	25.00	500	12.75	15.55	8.50
**4**	10.00	3.00	5.00	18.00	500	13.20	14.90	9.50
**5**	5.00	5.00	10.00	20.00	800	12.30	15.40	13.45
**6**	5.00	3.00	5.00	13.00	500	12.75	14.95	12.50
**7**	5.00	5.00	5.00	15.00	800	9.60	16.90	12.55
**8**	5.00	3.00	15.00	23.00	500	12.10	10.80	9.50
**9**	5.00	5.00	15.00	25.00	800	15.60	18.45	14.75
**10**	3.00	3.00	3.00	9.00	500	8.70	7.90	8.50
**11**	3.00	5.00	3.00	11.00	800	13.40	14.10	12.90
**12**	3.00	3.00	10.00	16.00	500	10.65	9.90	7.50
**13**	3.00	5.00	10.00	18.00	800	17.95	18.50	12.55
**14**	3.00	5.00	10.00	18.00	500	12.40	12.10	10.35

**Table 3 biology-09-00303-t003:** Electrodes for in vivo experiments. Details of the damaged parenchyma using the setting of Tests.

Test	Electrode	P1 (mm)	S (mm)	P2 (mm)	D (P1) (mm)	D (P2) (mm)	Voltage (V)	No. of Pulses	Parenchymal Damage (mm^2^)
**1**	**1**	5.00	3.00	20.00	1.80	2.00	800	80	9.18 × 10.03
**2**	**1**	5.00	3.00	20.00	1.80	2.00	1200	80	8.89 × 7.19
**3**	**2**	5.00	3.00	10.00	1.80	2.00	800	80	12.99 × 11.82
**4**	**2**	5.00	3.00	10.00	1.80	2.00	1200	80	8.06 × 9.56

**Table 4 biology-09-00303-t004:** Stainless-steel AISI 4340 properties.

	AISI 4340	Unit of Measure
Density	7850	kg/m^3^
Young’s modulus	205 × 10^9^	Pa
Thermal conductivity	44.5	W/mK
Specific heat	475	J/kgK
Electric conductivity	4.032 × 10^6^	S/m
Thermal expansion coefficient	12.3 × 10^−6^	K^−1^
Relative permittivity	1	-
Poisson’s ratio	0.28	-

**Note.** AISI = American Iron and Steel Institute.

**Table 5 biology-09-00303-t005:** Polyimide properties.

Properties	Polyimide	Unit of Measure
Density	1300	kg/m^3^
Young’s modulus	3.1 × 10^9^	Pa
Thermal conductivity	0.15	W/mK
Specific heat	1100	J/kgK
Electric conductivity	10^−10^	S/m
Relative permittivity	4	-
Poisson’s ratio	0.28	-

**Table 6 biology-09-00303-t006:** Animal liver properties. Example of the properties considered to simulate the tissue/organ in which to insert the device.

Properties	Animal Liver	Unit of Measure
Electric conductivity	0.07	S/m
Relative permittivity	80	-

**Table 7 biology-09-00303-t007:** Characterization of components used to make the electrode prototype.

	Outer Diameter (OD)	Inner Diameter (ID)	Wall Thickness
Stainless-steel mandrel	1.40 ± 0.02 mm	-	-
Reduced stainless-steel mandrel	0.83 ± 0.02 mm	-	-
First insulated sheath	0.9144 mm	0.8636 ± 0.010 mm	0.0254 ± 0.0064 mm
Second conductive pole	1.15 ± 0.02 mm	0.95 ± 0.02 mm	0.20
Second insulated sheath	1.3843 mm	1.1938 ± 0.013 mm	0.0953 ± 0.0239 mm

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
