# Peer review of "Design and Characterization of a Minimally Invasive Bipolar Electrode for Electroporation"

_biology, 2020, doi:10.3390/biology9090303_

Round 1

Reviewer 1 Report

Interesting paper. Some minor comments. 

  1. Many of the Comsol figures appear faded out and are difficult to read. These should be corrected.
  2. Not clear why the voltage was limited to 1500V when the max generator output can be much higher.
  3. 1 cm ablation size is not very impressive or clinically relevant for a single electrode. To compare, a single MWA probe can create a 3 cm dia lesion.
  4. The clinical application of an ablation with assymetric geometry is unclear 

Author Response

  1. Many of the Comsol figures appear faded out and are difficult to read. These should be corrected.

As suggested by reviewer we modified the Figures to improve quality and visibility.

  1. Not clear why the voltage was limited to 1500V when the max generator output can be much higher.

Actually, the maximum voltage of 1500 V is linked to the electric field (voltage on distance between two poles) and it is sufficient to reach in these experiment the irreversible electroporation threshold. Moreover, a higher voltage value should be avoided to prevent electric hazards (sparks, high current, etc.).

We added the previous sentence in the discussion section.

  1. 1 cm ablation size is not very impressive or clinically relevant for a single electrode. To compare, a single MWA probe can create a 3 cm dia lesion.

As suggested by Reviewrs we added the following paragraph in the discussion section:

The main goal of our study is to reach a trade off among the treatment invasiveness, linked to the probe diameter, and the electroporated area. Therefore, we tried to reduce as much as possible the diameter of the device but ensuring an electroporated area of at least of 10 mm of dimeter. Based on our knowledges, the conventional microwave (MWA) single probes diameter are much higher than 1.4 mm [44–47] and then reach ablated area greater than 10 mm. In fact as reported by Fallahi et al [45], although printed circuit board technologies such as microstrips provide the freedom to design antennas with diverse ablation patterns, achieving a practical device with an overall diameter less than 3 mm is not feasible at common MW ablation frequencies of 915 MHz and 2.45 GHz due to the large wavelength. In Colebeck et al. [47] a slot antenna was printed on the back side of a microstrip line with ultrawideband characteristic. The device is capable of creating ablation zones at 915 MHz, 2.45 GHz, and 5.8 GHz. However, the overall width of the device is 5.5 mm, making it unsuitable for most clinical MW ablation applications.

  1. The clinical application of an ablation with asymmetric geometry is unclear 

As suggested by reviewer we added the following sentence in the discussion section:

Asymmetric geometry has investigated in order to evaluate the capability of shaping the electroporated area. Thanks to the asymmetric geometry, in fact, it is possible to focus the electric field distribution around the shortest pole more than around the longest one. In this way we obtain a “drop” shaped electroporated volume, which can be used to treat specific cancer lesions, preserving tissue not intended to be treated such as the vertebral metastasis.

Reviewer 2 Report

In this study, the authors designed and characterized a minimally invasive bipolar electrode for reversible electroporation. This single needle electrode, as they claimed could improve the application of electroporation for treating deep lesions and with the size over 1 cm of diameter. Several problems should be solved before this study can be accepted.

  1. The author investigated the effect of the design of the bipolar electrode and applied voltage on the ultimate treating diameter. How about pulse number, pulse length, and pulse interval (pulse frequency)? Will these significantly affect the treating size?
  2. Figure. 5 looks like an illustration experimental report and makes me confused. Please sketch the electroporated area, add a clear scale bar, and rename the samples.
  3. Scale bar in Figure. 6 is not clear
  4. In both animal and vegetable tests, untreated samples are recommended to display.
  5. The unit electroporated volume should be consistent in this study. Notably, diameter and area are in mixed-use in this study.
  6. In test 5 and 6 of the second part of the theoretical study, the authors chose the voltage of 800 V, but not used an intermediate applied voltage value of 1000 V as claimed before. Why?
  7. As for the experimental design and statistics, parallel specimens and scientific size measures are needed.

Author Response

  1. The author investigated the effect of the design of the bipolar electrode and applied voltage on the ultimate treating diameter. How about pulse number, pulse length, and pulse interval (pulse frequency)? Will these significantly affect the treating size?

The aim of the study is to optimize the design of the electrode and no to optimize the electric protocol. As reported in the methods section, the electric protocol is fixed to reach the irreversible electroporation and consist of in a standard protocol: ten or twelve 8-pulses trains; 100 μs pulses length; 1 Hz pulse frequency.

  1. Please sketch the electroporated area, add a clear scale bar, and rename the samples.

            As suggested by reviewer we modified the Figures to improve quality and visibility.

  1. Scale bar in Figure. 6 is not clear

                  As suggested by reviewer we modified the Figure 6.

  1. In both animal and vegetable tests, untreated samples are recommended to display.

            As suggested we provided in the figure or in the caption indication about the untreated       samples.

  1. The unit electroporated volume should be consistent in this study. Notably, diameter and area are in mixed-use in this study.

            As suggested by reviewer, we reported all diameter and area in mm

  1. In test 5 and 6 of the second part of the theoretical study, the authors chose the voltage of 800 V, but not used an intermediate applied voltage value of 1000 V as claimed before. Why?

I apologized but we deleted the phrase because it was referred only to the test with symmetrical configuration. However, we integrated the test with asymmetrical configuration (test 5 and 6) with voltage values of 1000V and 1500V.

  1. As for the experimental design and statistics, parallel specimens and scientific size measures are needed.

As suggested by the reviewer, we have modified all measures in the text using two decimal places.

Reviewer 3 Report

Merola et. al. present a minimally invasive prototype electrode that can be used for electroporation. They present a computational analysis that is complimented by a "pre-clinical and "animal model".

Major and minor comments are as follows;

line 76 - "...... authors state new device is designed for geometry and design, but also conditions of use" - its not clear what this statement means. Please elaborate on how the geometry, design, and conditions the device presented here can be used for differ from current technologies

line 54 - "....to determine a reversible or irrevesible electroporation....."what does a reversible or irreversible electroporation mean? Electroporation itself is a process/technique. Are you talking about electroporation parameters, maybe?

lines 64 - 66 - this sentence seems out of place and somewhat odd. What treatment area are the authors talking about and why is there a protocol referenced. This reviewer suggests rewriting this sentence to make it more clear or placing it in a section more appropriate.

Figure 3 a,b,d,e,g,h,l,m its not clear what we are look at in the figures as the text is very small, what are your color plots representing? what is the scale, and what are the units of whatever value they are reporting.

how do the electrode diameters you all tried compare to what is currently used in the field? Are the smaller. larger, or within range? Please mention this point in the paper

For COMSOL model what computational elements did you use to model your system and how many elements did you use? Mesh size, etc. Also, when you applied your voltages in your model was the voltage static or cyclic?

line 181 - "black area,....observed 24 hours after treatment" How long were the potatoes  in general electroporated and why did you wait 24 hours later to image instead of immediately after the electroporation ended?

figure 5- not sure how the reader is supposed to tell the difference between each of these figures. Based off of the looks of the images I assume there are differences in voltages and geometry? Also, the section this figure is referenced in talks about symmetrical vs. asymmetrical geometry. Which images in this figure are symmetical vs. asymmetrical? Adding labels to these figures would be helpful.

line 208 - "......a diameter of 1 cm can be obtained,.... main purposes of this study" - why is 1 cm diameter a goal here? What's significant about this size?

line 210 - "...exceeds 8.9x7.2 mm^2" - this representation of the area is a bit odd. How do you have units of a 8.97 x 7.2 mm squared? Out of curiosity what shape are these dimensions supposed to represent the area of and how did you get this number? Please update this.

Figure 6 - I would suggest labeling in this figure explicitly the area that's ablated and healthy or alive, scale bar should be added to this figure as well.

authors should expand more on how their current technology is novel and compares to what currently exists. Advantages and disadvantages would be helpful as well

line 316 - "...has good mechanical properties" - what are these "good mechanical properties" please be more specific

line318 - again what does "excellent mechanical characteristics" mean

line 432 - "..according to standard protocol" what standard protocol was used? Please provide more detail or at least site a reference.

multiple typos and grammatical errors exist throughout the manuscript. Please proofread and correct. A few examples are listed below

line 57 - time should be "times"

line 58 - remove "the" before cell nerosis

line 58 - "......which can be successfully exploit...." - wrong tense please correct

line 61 - cell should be "cells' "

line 74 - trough, typo

line 101 - "....the volume increased of 5%...." grammatical error please correct

line 385 - "...electrode was totally inserted...." please use a different word than "totally". This reads a bit awkward in the sentence. Suggested to use a different word and/or rewrite sentence

line 427 - "..... a light microscopy...", maybe you mean a light microscope?

Author Response

Major and minor comments are as follows;

line 76 - "...... authors state new device is designed for geometry and design, but also conditions of use" - its not clear what this statement means. Please elaborate on how the geometry, design, and conditions the device presented here can be used for differ from current technologies

As suggested by reviewer we modified the paragraph as follows:

The aim of the study was to design and characterize a minimally invasive bipolar electrode for electroporation. The new device is different from the already marketed electrodes for its geometry, design and utilize; it consists of a single insertion bipolar electrode, where by the anode and cathode components are contained on a single needle, it have been explored as an approach to reduce the number of electrode insertions required which may substantially simplify the electrode placement for the procedure, minimize invasiveness and saving time. In fact the already marketed electrodes that involves the use of two different needles as opposite poles and the correct placement of themselves.

The device was realized to improve and facilitate the application of electroporation, and is aimed to be connected to the electroporator device which controls the generation and delivery of electrical impulses and evaluates the procedure outcome. The intended purpose of the new electrode is to reach a trade off among the treatment invasiveness, linked to the probe diameter, and the electroporated area. Therefore, we tried to reduce as much as possible the diameter of the device but ensuring an electroporated area of at least of 10 mm of dimeter.

line 54 - "....to determine a reversible or irreversible electroporation....."what does a reversible or irreversible electroporation mean? Electroporation itself is a process/technique. Are you talking about electroporation parameters, maybe?

As suggested by reviewer we modified the phrase as follows:

As is known, it is possible to use different electroporation parameters (number and voltage amplitude of pulses) in order to obtain reversible or an irreversible effect on the electroporated cells membranes.

lines 64 - 66 - this sentence seems out of place and somewhat odd. What treatment area are the authors talking about and why is there a protocol referenced. This reviewer suggests rewriting this sentence to make it more clear or placing it in a section more appropriate.

As suggested by reviewer, we deleted this sentence.

Figure 3 a,b,d,e,g,h,l,m its not clear what we are look at in the figures as the text is very small, what are your color plots representing? what is the scale, and what are the units of whatever value they are reporting.

As suggested by reviewer we modified the Comsol figures and legends, we dived the figure in two different panel to improve quality and visibility.

how do the electrode diameters you all tried compare to what is currently used in the field? Are the smaller. larger, or within range? Please mention this point in the paper

As suggested by reviewer we inserted in the text the following sentence:

The chosen electrode diameter is within range of already commercialized needle (VGD needles series of Igea SpA, Carpi-Modena, Italy) that vary among 0.8-1.8 mm.

For COMSOL model what computational elements did you use to model your system and how many elements did you use? Mesh size, etc. Also, when you applied your voltages in your model was the voltage static or cyclic?

As suggested by reviewer we added the following sentence in the Comsol Multiphysics section.

We have set the voltage applied as static and the mesh as unstructured with tetrahedral elements.  The computation elements used to recreate the model are: full cylinders for two conductive poles (P1 and P2) and the insulating pole (S).

line 181 - "black area,....observed 24 hours after treatment" How long were the potatoes  in general electroporated and why did you wait 24 hours later to image instead of immediately after the electroporation ended?

As suggested by reviewer we added the following sentence in the Vegetable model results section.

The timing of 24h was chosen to better visualize the ablated area, not visible immediately after the electroporation ended.

figure 5- not sure how the reader is supposed to tell the difference between each of these figures. Based off of the looks of the images I assume there are differences in voltages and geometry? Also, the section this figure is referenced in talks about symmetrical vs. asymmetrical geometry. Which images in this figure are symmetical vs. asymmetrical? Adding labels to these figures would be helpful.

As suggested by reviewer we modify the Figure and the caption related to vegetable test.

line 208 - "......a diameter of 1 cm can be obtained,.... main purposes of this study" - why is 1 cm diameter a goal here? What's significant about this size?

As suggested by reviewer we delete this sentence related to the size.

The purpose of this device is to facilitate operating procedures for treating deep lesions by reducing the invasiveness of treatment and allowing treatment of a spherical area of at least 10 mm of diameter.

line 210 - "...exceeds 8.9x7.2 mm^2" - this representation of the area is a bit odd. How do you have units of a 8.97 x 7.2 mm squared? Out of curiosity what shape are these dimensions supposed to represent the area of and how did you get this number? Please update this.

As suggested by reviewer and for major understanding, we reported also for the experimental animal test the maximum diameter if the ablated area.

Figure 6 - I would suggest labeling in this figure explicitly the area that's ablated and healthy or alive, scale bar should be added to this figure as well.

As suggested by reviewer we modified the Figure 6 and the caption.

authors should expand more on how their current technology is novel and compares to what currently exists. Advantages and disadvantages would be helpful as well

As suggested by reviewer, the discussion section was revised to expand the advantages of proposed technology compared with the already commercialized products for ECT, IRE and MWA.

line 316 - "...has good mechanical properties" - what are these "good mechanical properties" please be more specific

As suggested by reviewer we inserted the following specification:

Austenitic stainless steel AISI 304 was selected for the electrode conductive poles because this material has good mechanical properties (tensile strength, yield strength, high corrosion resistance) compared to ferritic and martensitic steels.

line318 - again what does "excellent mechanical characteristics" mean

As suggested by reviewer we rewritten the phrase as follows: For the insulating pole, the choice of materials fell between polyimide and polyethylene terephthalate being among the most used polymers in the biomedical field. Both are thermoplastic polymers. Polyimide (PI) was chosen for its coefficient of friction, good vibration-damping properties, abrasion resistance and high dielectric strength.

line 432 - "..according to standard protocol" what standard protocol was used? Please provide more detail or at least site a reference.

As required by reviewer we added the reference 44 for the caspase-3 method.

multiple typos and grammatical errors exist throughout the manuscript. Please proofread and correct.

As required by reviewer we revised all manuscript and we corrected the typos and grammatical errors.

A few examples are listed below

line 57 - time should be "times"

done

line 58 - remove "the" before cell nerosis

done

line 58 - "......which can be successfully exploit...." - wrong tense please correct

done

line 61 - cell should be "cells' "

done

line 74 - trough, typo

done

line 101 - "....the volume increased of 5%...." grammatical error please correct

done

line 385 - "...electrode was totally inserted...." please use a different word than "totally". This reads a bit awkward in the sentence. Suggested to use a different word and/or rewrite sentence

done

line 427 - "..... a light microscopy...", maybe you mean a light microscope

done

Round 2

Reviewer 2 Report

The manuscript can be accepted in the present form.

Author Response

We revised the entire article to improve english and delete typos. 

Reviewer 3 Report

Authors have addressed all comments sufficiently.

Author Response

(The authors gave the same response as above.)
